# Educational Effects of Simulation and Non-Simulation Training in Airway Management according to Levels of the Kirkpatrick Model: A Systematic Review and Network Meta-Analysis

**DOI:** 10.3390/jcm11195614

**Published:** 2022-09-23

**Authors:** Koichi Ando, Akane Ando, Akihiko Tanaka, Shinji Koba, Hironori Sagara

**Affiliations:** 1Division of Respiratory Medicine and Allergology, Department of Medicine, Showa University School of Medicine, 1-5-8 Hatanodai, Shinagawa-ku, Tokyo 142-8666, Japan; 2Division of Internal Medicine, Showa University Dental Hospital Medical Clinic, Senzoku Campus, Showa University, 2-1-1 Kita-senzoku, Ohta-ku, Tokyo 145-8515, Japan; 3Department of Anesthesiology, Showa University School of Medicine, 1-5-8 Hatanodai, Shinagawa-ku, Tokyo 142-8666, Japan; 4Division of Cardiology, Department of Medicine, Showa University School of Medicine, 1-5-8 Hatanodai, Shinagawa-ku, Tokyo 142-8666, Japan

**Keywords:** simulation training, airway management, systematic review, network meta-analysis

## Abstract

There is insufficient validation of the effectiveness of simulation-based training (Sim) or non-simulation-based training (non-Sim) for teaching airway management to healthcare professionals within the literature. We thus conducted a network meta-analysis comparing the effectiveness of Sim, non-Sim, and no educational intervention (NI) for airway management. The primary endpoints were knowledge scores (KnS) and behavioral performance scores (BpS) corresponding to assessments at levels 2 and 3 of the Kirkpatrick model, respectively. Effect sizes were expressed as standardized mean differences (Std. MD) and 95% credible intervals (CrIs). Regarding KnS, the educational effects of Sim and non-Sim were significantly improved compared to those of NI (Std. MD [95% CI]: 1.110 [0.903–1.316] and 0.819 [0.209–1.429], respectively); there was no significant difference between Sim and non-Sim. The educational effect of Sim in BpS was significantly improved compared to that of non-Sim and NI (0.850 [0.015–1.691] and 0.660 [0.241–1.076]); there were no differences between non-Sim and NI. Surface under the cumulative rank curve values demonstrated that Sim ranked highest in efficacy for KnS and BpS. This study provides valuable information regarding the educational efficacy of Sim and non-Sim in airway management. Larger randomized controlled trials are needed to confirm these findings.

## 1. Introduction

In acute care settings, failure of proper airway management is strongly associated with increased mortality [1]. Therefore, effective airway management training for medical staff has become an important issue within clinical education [2,3,4]. However, optimal strategies for education and training methods directly related to the improvement of airway management skills and patient outcomes have not yet been fully standardized [5]. In recent years, the importance of simulation-based training (Sim) in improving medical skill proficiency has been reported in several studies [6,7]. The major advantage of Sim is that it is expected to improve knowledge and skills in an environment where there is no realistic risk of harm to the patient [6]. 

Previous randomized controlled trials (RCTs) comparing the effects of Sim with those of non-simulation-based training (non-Sim) in airway management [8,9,10,11,12] have shown inconsistent results, whereas previous meta-analyses comparing Sim with non-Sim or with no training have highlighted the advantages of Sim [13,14]. However, many previous studies have been limited in their evaluation due to small sample sizes. Furthermore, to our knowledge, no clear ranking of these training methods has been completed previously [13,14]. Nevertheless, Sim in airway management has been adopted by many healthcare organizations and is expected to become increasingly popular in the coming years [6,7,15]. However, the coronavirus disease 2019 (COVID-19) pandemic has affected various educational settings and the availability of in-person instruction, including medical education [16,17,18,19,20,21,22,23,24,25,26,27,28]. Sim conducted face to face is often withheld to prevent the spread of infection between instructors and learners or between learners. In such cases, non-Sim, which also allows for providing individualized instruction, should be included in clinical education as an appropriate alternative [29]. Therefore, information on the evaluation of non-Sim in comparison to that of Sim and no educational intervention (NI) is necessary. 

To our knowledge, there are few reports comparing the efficacy of non-Sim and NI in airway management education. In addition, there is still no consensus on the detailed efficacy profiles of Sim and non-Sim in airway management, and there are no standardized, uniform guidelines for airway management training. Strong evidence on airway management training is needed in order for educators to select appropriate teaching methods. Therefore, from the perspective of medical education, it is essential to evaluate the effectiveness and ranking of training methods, including Sim and non-Sim, in airway management according to Kirkpatrick’s educational strategy evaluation model [30]. Specifically, by using network meta-analysis statistical methodology, it is possible not only to compare the effects of multiple intervention methods simultaneously, but also to rank the effects of each intervention method [31,32]. The purpose of this study was to compare and rank the effects of Sim, non-Sim, and no training on airway management knowledge and skills using the Kirkpatrick model and Bayesian network meta-analysis.

## 2. Materials and Methods

### 2.1. Systematic Review

We conducted a comprehensive literature search in order to identify RCTs and observational studies published since 1946 regarding the effects of Sim on airway management among healthcare workers. The reason why the literature search covers the period after 1946 is because this is the period that can be searched by the databases searched. In screening the initial literature search, the time period covered was as broad as possible to avoid the risk of missing references that would be eligible for inclusion. Four databases (PubMed, Cochrane Library, EMBASE, and SCOPUS) were searched on 16 August 2021. Keywords including airway management, simulation training, and education as well as their medical subject headings were used to construct the search strategy. For example, in the PubMed database, ((training [Title/Abstract]) OR (education [Title/Abstract])) AND ((simulation [Title/Abstract]) AND (airway management [Title/Abstract])) were the selected search terms. In addition, the reference lists of the identified studies were checked in order to avoid the risk of missing relevant studies that met the inclusion criteria. If we could not obtain sufficient information from the data reported in a particular study, we contacted the corresponding author of that study via e-mail or utilized the data reported in previous systematic reviews. 

This systematic review (registration: UMIN-CTR no. 000039454) aimed to identify all clinical studies on the effects of Sim in airway management education. This study was conducted in accordance with the PRISMA (Priority Reporting for Systematic Review and Meta-Analysis) Statement [33] and the PRISMA Extension Statement for Network Meta-Analysis [34] guidelines. Two researchers (KA and AA) conducted the literature search independently. Study inclusion and exclusion criteria were applied using a predefined patient, intervention, comparison, outcome, and study design (PICOS) approach. In addition, clinical and methodological heterogeneity were addressed and the validity of indirect comparisons was assessed according to standard methodology.

### 2.2. Inclusion Criteria

Studies were included if the protocol met the following criterion: Evaluation of the effectiveness of Sim in the airway management training of healthcare professionals. Studies not evaluating Sim were excluded from the current meta-analysis and systematic review.

We defined simulation education of airway management as an educational tool that includes not only direct laryngoscopy (DL) intubation for emergency laryngoscopy, but also fiberoptic intubation (FOI), supraglottic airway management (laryngeal mask airway [LMA] or combi tube)

### 2.3. Interventions/Comparisons

We included studies that evaluated either Sim or non-Sim within airway management education in this network meta-analysis (NMA). The definition of NI group indicates a group in which no pedagogical intervention, either simulation or non-simulation education, is provided (e.g., a group that only did self-study). Pre- and post-comparison studies represent the preliminary stage of a pedagogical intervention.

### 2.4. Outcomes

The primary efficacy endpoints assessed in this study were knowledge scores (KnS) and behavioral performance scores (BpS), expressed as standardized mean differences (Std. MD) and 95% credible intervals (CrI). In addition, the surface under the cumulative rank curve (SUCRA) values for KnS and BpS were calculated in order to rank the effectiveness of each training method in airway management education [35]. These predefined endpoints were analyzed only when relevant data were available; two authors (KA and AA) independently extracted the relevant data.

### 2.5. Study Design

We included RCTs and observational studies of Sim or non-Sim for airway management with at least one preset efficacy endpoint in this NMA.

### 2.6. Statistical Analysis

We used Bayesian NMA in order to assess the effectiveness of Sim in airway management education using predefined efficacy endpoints according to established methodologies outlined by the National Institute for Health and Care [36,37]. This is an established statistical NMA method and is recommended by the International Society for Pharmacoeconomics and Outcome Research guidelines for indirect comparisons and NMA, as well as by the National Institute for Health and Clinical Excellence (UK) and the Haute Autorité de Santé (France) [36,37]. 

In our analysis, we used the standard NMA method described by Dias et al. [38,39,40]. Further, we used a Bayesian model that assumed heterogeneity among the included studies with a non-informative prior distribution and Gibbs sampling using Markov chain Monte Carlo methodology in order to estimate the posterior treatment effect distribution [31,41]. We performed 50,000 iterations, with the first 10,000 iterations considered as burn-in samples. Model convergence was evaluated using Brooks-Gelman-Rubin (BGR) diagnostics [41,42]. The treatment effects were expressed as Std. MDs and 95% CrIs. The CrIs were derived based on the 2.5th and 97.5th percentiles of the posterior distribution. Results were considered statistically insignificant if the 95% CrI crossed the invalid line (i.e., Std. MD = 0). 

NMA allows for both the comparison and ranking of treatment groups. In this NMA, we ranked treatments based on SUCRA values calculated through Bayesian analysis [35]; SUCRA values range from 0 to 100%, with higher values indicating greater treatment effectiveness. A 100% value indicates the most ideal treatment [35]. We performed the analysis using OpenBUGS 1.4.0 software (MRC Biostatistics Unit, Cambridge Public Health Research Institute, Cambridge, UK) and we used STATA (v14, StataCorp., College Station, TX, USA) to create graphics for displaying the results.

### 2.7. Ethical Review

Due to the nature of this study, which is a systematic review and meta-analysis, ethical review board approval and patient consent were formally waived by the ethics review board at our institution.

## 3. Results

### 3.1. Systematic Review

This systemic literature review identified 266, 375, 152, and 144 studies from the PubMed, EMBASE, CENTRAL, and SCOPUS databases, respectively; among them, 719 articles remained after duplicate removal. After applying the PICOS approach, 18 studies were retained. Figure 1 shows the study selection process, and Appendix A shows the characteristics of the included studies.

We used data from 12 [12,43,44,45,46,47,48,49,50,51,52,53] and 6 [54,55,56,57,58,59] reports, respectively, to evaluate the KnS and BpS. In all the analyses, we applied BGR diagnostics and assessed model convergence [41,42]. We confirmed favorable model convergence in all analyses. Figure 2 shows the network map of this NMA.

### 3.2. Primary Efficacy Endpoint: Knowledge Score

We found that the KnS was statistically significantly improved in the Sim and non-Sim groups compared to that in the NI group (Std. MD [95% CrI] = 1.110 [0.903 to 1.316] and 0.819 [0.209 to 1.429], respectively). There were no statistically significant differences between the Sim-educated and non-Sim-educated groups, nor between the Sim and non-Sim groups (Std.MD [95% CrI] = 0.290 [−0.279 to 0.865]).

The SUCRA values for the KnS for Sim, non-Sim, and NI were 92.0%, 57.8%, and 0.2%, respectively (Figure 3, Table 1).

### 3.3. Primary Efficacy Endpoint: Behavior Performance

BpS was statistically significantly improved in the Sim group compared with that in the non-Sim and NI groups (Std.MD [95% CrI] = 0.850 [0.015 to 1.691 and 0.660 [0.241 to 1.076], respectively) (Figure 4). There were no statistically significant differences in BpS between the non-Sim and NI groups (Std. MD [95% CI] = −0.192 [−1.130 to 0.747]). The SUCRA values for the time skill component of Sim, non-Sim, and no intervention were 98.8%, 18.4%, and 32.8%, respectively (Table 2).

## 4. Discussion

In this study, we adopted an NMA statistical methodology in order to compare and rank educational effectiveness in airway management education among Sim, non-sim, and NI groups at levels 2 and 3 of the Kirkpatrick model. For BpS (corresponding to level 3 of the Kirkpatrick model), we observed that Sim was statistically significantly more effective than non-Sim and NI and that there was no statistically significant difference between the Sim and non-Sim groups in terms of educational efficacy. For BpS (corresponding to level 3 of the Kirkpatrick model), Sim had a statistically significantly improved educational effect as compared to non-Sim and NI, while no statistically significant differences in educational effect were found between the non-Sim and NI groups.

Previous studies have validated the effectiveness of Sim [8,9,10,11,12,60], but most of these studies had small sample sizes and the results were inconsistent. In addition, although previous meta-analyses compared only two groups, such as Sim and NI or Sim and non-Sim, using traditional pairwise meta-analysis, these studies did not rank the treatment groups, including Sim, non-Sim, and NI [13,14]. Furthermore, previous systematic reviews have discussed the effectiveness of airway management education according to the level of the Kirkpatrick model [61]. However, prior to the present study, the comparison between non-Sim and NI had not been performed. To our knowledge, our study is the first NMA to employ knowledge and skills as outcomes for assessment at levels 2 and 3 of the Kirkpatrick model and to simultaneously compare each pair of the three treatment arms, namely Sim, non-Sim, and NI, in order to rank the effectiveness of each treatment arm for airway management education. 

In addition, during the ongoing COVID-19 pandemic, more attention needs to be paid to infection control. One of the main conclusions of our paper is that, at Kirkpatrick level 2, no significant differences were shown between Sim group and non-Sim groups according to the results of the ranking evaluation, although simulation education had the most favorable educational effects at both Kirkpatrick levels 2 and 3. non-simulation education can adopt a more infection control-friendly format compared to simulation education. As an alternative to face-to-face simulation education, non-simulation education with more attention to infection control may be used. In other words, our study suggests that appropriate use of simulation and non-simulation education may be able to achieve both educational effectiveness and infection control.

However, whether such non-simulation education can be used as an alternative to simulation education has not been sufficiently discussed so far. Therefore, it is important and necessary to compare between non-Sim and NI from the viewpoint of infection control. The present study was conducted because there was a need for data comparing the educational effectiveness of non-Sim with that of Sim and NI at each level of the Kirkpatrick model. 

The results of the present study revealed that that at level 2 of the Kirkpatrick model (i.e., knowledge scores), both Sim and non-Sim had a significantly improved effect on airway management education compared with NI, and there were no significant differences in educational effectiveness between Sim and non-Sim. At level 3 (i.e., skills), Sim was significantly more effective for airway management education compared with non-Sim and NI, while there were no significant differences in educational effectiveness between non-Sim and NI. These results provide valuable insights into the effectiveness of non-Sim and Sim in airway management education. 

These results can be explained by educational theory. In particular, Sim is medically safe for both learners and patients because learning takes place in a simulated environment, and it can effectively train medical professionals in clinical skills, including physical examination [7,61]. In addition, training in actual clinical settings forces learners to learn while practicing patient-centered medicine in complex situations that vary depending on the patient’s condition and the number and capabilities of medical staff. Therefore, both learners and instructors may be exposed to stress during in-person learning in clinical settings [7]. However, in Sim, instructors can plan and implement customized learning based on learners’ knowledge and technical readiness [4]. Thus, Sim allows for the development of learner-centered training that is somewhat detached from the clinical setting. This educational feature of Sim seems to be particularly useful in the field of airway management training, as it is necessary to acquire not only the skills needed as a medical professional but also the non-technical skills needed in a medical team, such as communication, leadership, and professionalism.

However, during the COVID-19 pandemic, the risk of spread of infection not only between instructors and learners but also between learners must be considered. In the present study, there was no significant difference in teaching effectiveness between Sim and non-Sim, especially at level 2 of the Kirkpatrick model, and non-Sim was significantly better for airway management education than NI. This result suggests that it is reasonable to understand the characteristics of non-Sim educational modalities and to effectively incorporate non-Sim into airway management education from the perspective of infection control. Our result shown that that there is no significant difference between Sim and non-Sim in the knowledge score (Kirkpatrick level 2). This suggests that replacing a part of medical education with non-Sim, while maintaining good educational effect, may be possible to achieve both educational effect and infection prevention measures. However, the accumulation of medical pedagogical evidence necessary to construct a specific curriculum for this purpose is still insufficient, and future research is desirable.

There are several limitations to this study. First, there are various means of classifying Sims (e.g., biological models, precision instruments for advanced skills, and mannequins for low-level skills) [15]. The fidelity of simulation instruments is not always consistent across the included studies. We cannot rule out the possibility that this may have influenced our final conclusions of this study. Similarly, non-Sim includes videos, problem-solving discussions, and self-study. Depending on the methodology and equipment used to teach, the effectiveness of learning may vary. However, the diversity of Sim and non-Sim interventions was not considered in this study. Therefore, it was not possible to determine the most desirable methodology among the various Sim and non-Sim interventions, and this needs to be verified in future studies. Second, the current study did not examine the cost of the interventions; Sim is generally expensive, but its cost depends on the specific equipment and methods implemented in a particular intervention [6,7]. Adequate validation is needed to determine whether a reasonable cost–benefit balance can be achieved, and future studies should examine the most appropriate Sim method in terms of cost-effectiveness. Finally, this study examined the effectiveness of airway management education according to the levels of the Kirkpatrick model, but the current evaluation was limited to levels 2 and 3 of the mode, and did not assess levels 1 (i.e., learner response) and 4 (i.e., patient outcome). It is recommended that these levels also be evaluated in future studies.

## 5. Conclusions

In summary, this study adopted NMA statistical methodology in order to compare and rank educational effectiveness in airway management education among Sim, non-Sim, and NI based on the Kirkpatrick model. Evaluations at level 2 of the Kirkpatrick model showed that both Sim and non-Sim were more effective than NI in terms of KnS, and that there was no corresponding statistically significant difference between Sim and non-Sim. In the evaluation at level 3 of the Kirkpatrick model, Sim was statistically significantly more effective than non-Sim and NI in terms of BpS, while there was no corresponding statistically significant difference in educational effectiveness between non-Sim and NI. These results provide important information on the educational effectiveness of Sim and non-Sim in airway management education. Further validation of the educational effectiveness of Sim and non-Sim is desirable for the establishment of a more effective airway management education system that simultaneously takes into account cost-effectiveness and infection control.

## Figures and Tables

**Figure 1 jcm-11-05614-f001:**
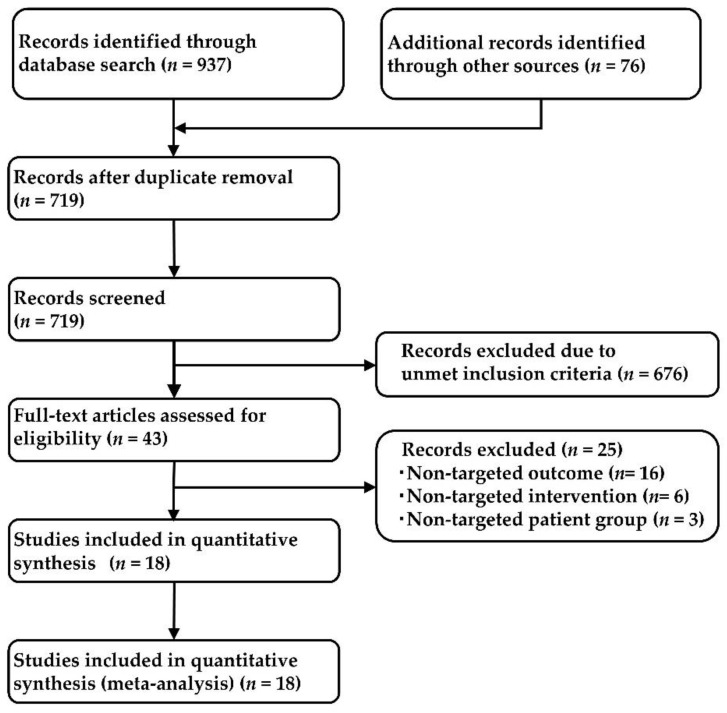
Flow diagram of the study selection process.

**Figure 2 jcm-11-05614-f002:**
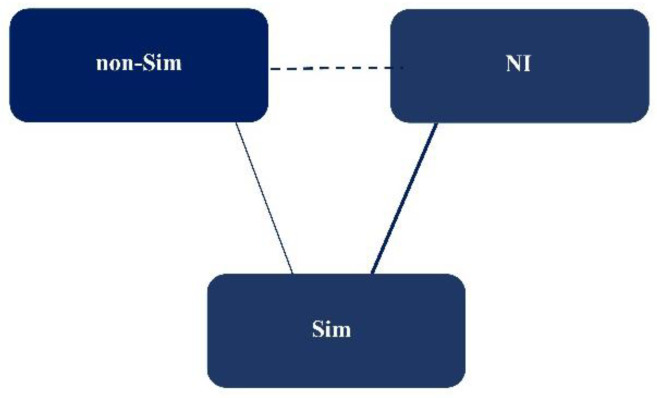
A network map showing the overview of the network meta-analysis performed in this study. The number of studies included in this analysis is indicated by the solid line (the number of studies is indicated by the thickness of the line). Dashed lines indicate relationships for which no randomized controlled trials (RCTs) were available, but for which indirect comparisons were attempted; Sim, simulation-based training; non-Sim, non-simulation-based training; NI, no educational intervention.

**Figure 3 jcm-11-05614-f003:**
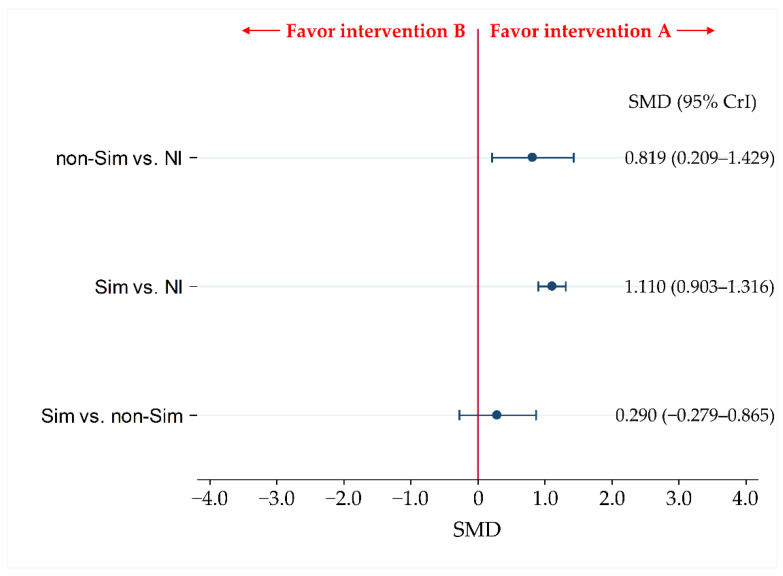
Comparison of the efficacy of knowledge scores for Sim, non-Sim, and no intervention. We expressed comparisons as intervention A vs. intervention B. Data were expressed as standardized mean differences (Std. MD) and 95% credible intervals (CrIs). Sim, simulation-based training; non-Sim, non-simulation-based training; NI, no educational intervention.

**Figure 4 jcm-11-05614-f004:**
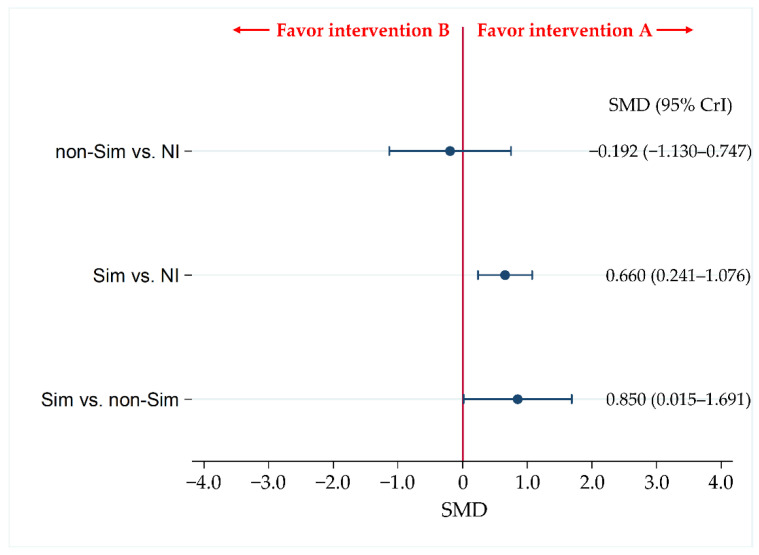
Comparison of efficacy for behavior performance of Sim, non-Sim, and no intervention. We expressed the comparisons as intervention A vs. intervention B. Data were expressed as standardized mean differences (Std. MD) and 95% credible intervals (CrIs). Sim, simulation-based training; non-Sim, non-simulation-based training; NI, no intervention.

**Table 1 jcm-11-05614-t001:** Surface under the cumulative ranking (SUCRA) curve values and ranking of efficacy for knowledge scores.

Intervention	Efficacy for Knowledge Scores, % (Rank)
Sim	92.0 (1)
Non-Sim	57.8 (2)
NI	0.2 (3)

Sim, simulation-based training; non-Sim, non-simulation-based training; NI, no educational intervention.

**Table 2 jcm-11-05614-t002:** Surface under the cumulative ranking (SUCRA) curve values and ranking of efficacy for behavior performance.

Intervention	Efficacy for Behavior Performance, % (Rank)
Sim	98.8 (1)
Non-Sim	18.4 (3)
NI	32.8 (2)

Sim, simulation-based training; non-Sim, non-simulation-based training; NI, no educational intervention.

## Data Availability

The authors confirm that the data sets analyzed in the present study, though not available in a public repository, will be made available from the corresponding author to other researchers upon reasonable request.

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
