# Peer review of "Educational Effects of Simulation and Non-Simulation Training in Airway Management according to Levels of the Kirkpatrick Model: A Systematic Review and Network Meta-Analysis"

_jcm, 2022, doi:10.3390/jcm11195614_

Round 1

Reviewer 1 Report

Thank you for giving me chance to read you manuscript.

The manuscript is well written and the results is clear.

This reviewer only suggest minor changes.

The authors should define the 'airway management' because this idea may differ between internal medicine and critical care medicine.

Author Response

Dear Reviewer 1

Thank you for giving us the opportunity to resubmit to Journal of Clinical Medicine a revised draft of our manuscript titled “Educational Effects of Simulation and Non-Simulation Training in Airway Management According to Levels of the Kirkpatrick Model: A Systematic Review and Network Meta-Analysis” (Manuscript ID: jcm-1888044). We sincerely appreciate the time and effort that you and the reviewers have dedicated to providing your valuable feedback on our manuscript. We are grateful to the reviewers for their insightful comments, and have been able to incorporate changes to reflect most of their suggestions. We have highlighted in yellow the changes within the manuscript.

Here is a point-by-point response to the reviewers’ comments and concerns.

Comment: Thank you for giving me chance to read you manuscript. The manuscript is well written and the results is clear. This reviewer only suggests minor changes. The authors should define the 'airway management' because this idea may differ between internal medicine and critical care medicine.

Response: We sincerely appreciate you pointing out this very important point. We agree with the reviewer’s comment. We defined simulation education of airway management as an educational tool that includes not only direct laryngoscopy (DL) intubation for emergency laryngoscopy, but also fiberoptic intubation (FOI), supraglottic airway management (laryngeal mask airway [LMA] or combi tube), and others. We have added an addendum to the main manuscript regarding this.

We are confident that our revised manuscript will be suitable for publication in Journal of Clinical Medicine and look forward to receiving your editorial decision.

Thank you for your consideration.

Sincerely,

Koichi Ando

Department of Medicine, Division of Respiratory Medicine and Allergology, Showa University School of Medicine

1-5-8 Hatanodai, Shinagawa-ku, Tokyo, 142-8666, Japan

Tel: +81-3-3784-8532

Fax: +81-3-3784-8742

Email: koichi-a@med.showa-u.ac.jp

Reviewer 2 Report

Thank you for the opportunity to review this manuscript.

The manuscript is well written, comprehensive, organized, and easy to follow. It appears that the results could apply to all healthcare professional students/education. Limitations are included. 

Figure 5 is not necessary. A brief description (i.e., the description provided with the figure) is adequate for the audience readers. 

Author Response

Dear Reviewer 2

Thank you for giving us the opportunity to resubmit to Journal of Clinical Medicine a revised draft of our manuscript titled “Educational Effects of Simulation and Non-Simulation Training in Airway Management According to Levels of the Kirkpatrick Model: A Systematic Review and Network Meta-Analysis” (Manuscript ID: jcm-1888044). We sincerely appreciate the time and effort that you and the reviewers have dedicated to providing your valuable feedback on our manuscript. We are grateful to the reviewers for their insightful comments, and have been able to incorporate changes to reflect most of their suggestions. We have highlighted in yellow the changes within the manuscript.

Here is a point-by-point response to the reviewers’ comments and concerns.

Comment: The manuscript is well written, comprehensive, organized, and easy to follow. It appears that the results could apply to all healthcare professional students/education. Limitations are included.

Figure 5 is not necessary. A brief description (i.e., the description provided with the figure) is adequate for the audience readers.

Response: I appreciate your very valuable comments and agree with the reviewer's comments, Figure 5 has been removed and a brief explanation incorporated.

We are confident that our revised manuscript will be suitable for publication in Journal of Clinical Medicine and look forward to receiving your editorial decision.

Thank you for your consideration.

Sincerely,

Koichi Ando

Department of Medicine, Division of Respiratory Medicine and Allergology, Showa University School of Medicine

1-5-8 Hatanodai, Shinagawa-ku, Tokyo, 142-8666, Japan

Tel: +81-3-3784-8532

Fax: +81-3-3784-8742

Email: koichi-a@med.showa-u.ac.jp

Reviewer 3 Report

It was a pleasure to analysis this review about simulation and airway management teaching. The idea to discriminate technical skills and knowledge is relativelly novel and merits to be. The choice of articles is correct and the manuscript is well-written. The bibliography is clear and well organized. The main conclusion is not a surprise but it is structured by statistics.

Nevertheless, I have some concerns and I hope the authors would be able to give me some explanations.

Why did you choose a so long period of inclusion. This makes no sense in simulation to start after the WWII! The last 20years are largely enough to obtain some homogeneity in the analysis. This is largely supported by the fact that techniques to achieve airway managements have changed (position, oxygenation, videolaryngoscopy...)

Second, I don't understand the 3d group: no educationnal! This is not clear and if you want to maintain this, you have to define the non interventional group and the population of no educational intervention: are they anesthesiologists, doctors?? This may impair the statistics and the message with no clear relevancy. I think that a table with the selected studies and their main results is missing to confirm the final results.

Third, there is a difference between high fidelity simulation and low-fidelity simulation and this is not described in your manuscript. I suppose that the main studies addressed on low-fidelity but I have no indication.

 To finish, you can mention the COVID period but the illustration number 5 is out of scope of the real question. 

You may discuss the best way to teach residents and maintain skills for senior anesthesiologists in regard to yours results.    Best Regards 

Author Response

Dear Reviewer 3

Thank you for giving us the opportunity to resubmit to Journal of Clinical Medicine a revised draft of our manuscript titled “Educational Effects of Simulation and Non-Simulation Training in Airway Management According to Levels of the Kirkpatrick Model: A Systematic Review and Network Meta-Analysis” (Manuscript ID: jcm-1888044). We sincerely appreciate the time and effort that you and the reviewers have dedicated to providing your valuable feedback on our manuscript. We are grateful to the reviewers for their insightful comments, and have been able to incorporate changes to reflect most of their suggestions. We have highlighted in yellow the changes within the manuscript.

Here is a point-by-point response to the reviewers’ comments and concerns.

It was a pleasure to analysis this review about simulation and airway management teaching. The idea to discriminate technical skills and knowledge is relatively novel and merits to be. The choice of articles is correct and the manuscript is well-written. The bibliography is clear and well organized. The main conclusion is not a surprise but it is structured by statistics.

Nevertheless, I have some concerns and I hope the authors would be able to give me some explanations.

Comment1: Why did you choose a so long period of inclusion. This makes no sense in simulation to start after the WWII! The last 20years are largely enough to obtain some homogeneity in the analysis. This is largely supported by the fact that techniques to achieve airway managements have changed (position, oxygenation, video laryngoscopy.)

Response1: Thank you very much for raising this very important point. The reason why the literature search covers the period after 1946 is because this is the period that can be searched by the databases searched. In screening the initial literature search, the time period covered was as broad as possible to avoid the risk of missing references that would be eligible for inclusion. In addition, the earliest simulation education in medicine was training for residents at a clinic in the United States in the early 1900s, and it was reported that the competence of residents improved after the training (Ellis et al 2003). In the 1950s, anesthesiologist Safar and his colleagues developed a simulator for cardiopulmonary resuscitation.35) In the 1950s, anesthesiologist Safar et al. developed a simulator for CPR (Safar et al. 1961). These reports support the validity of our assumption that the period of our literature search for this systematic review was after 1946.

Comment2: Second, I don't understand the 3d group: no educational! This is not clear and if you want to maintain this, you have to define the non-interventional group and the population of no educational intervention: are they anesthesiologists, doctors?? This may impair the statistics and the message with no clear relevancy. I think that a table with the selected studies and their main results is missing to confirm the final results.

Response2: Thank you for pointing this out to us. The points you raised are very important. We agree with the reviewer’s comment. The definition of no intervention indicates a group in which no pedagogical intervention, either simulation or non-simulation education, is provided (e.g., a group that only did self-study). Pre- and post-comparison studies represent the preliminary stage of a pedagogical intervention. We have added an addendum to the text regarding this. While it was difficult to summarize all of the key results of the incorporated studies in the table, the following comments have been added to the footnotes of TableS1 for the sake of reproducibility. “In cases where the inclusion studies did not report sufficient data for analysis, the data reported in reference number 13 (which was not included in the inclusion studies) were used in the integrated analysis.”

Comment3: Third, there is a difference between high fidelity simulation and low-fidelity simulation and this is not described in your manuscript. I suppose that the main studies addressed on low-fidelity but I have no indication.

Response3: Thank you for pointing out this very important point. We agree with the reviewer’s comment. As you have pointed out, the fidelity of simulation instruments is not always consistent across the included studies. We cannot rule out the possibility that this may have influenced our final conclusions of this study. This is a serious limitation of this report, and we have again noted it in the limitation section.

Comment4: To finish, you can mention the COVID period but the illustration number 5 is out of scope of the real question.

Response4: Thank you for pointing this out. We agree with the reviewer's remarks. In accordance with other reviewers' comments, Figure 5 was deleted and a brief explanation was added as follows; one of the main conclusions of our paper is that, according to the results of the ranking evaluation, simulation education had the most favorable educational effects at both Kirkpatrick levels 2 and 3. However, at Kirkpatrick level 2, no significant differences were shown between the simulation and non-simulation education groups. Non-simulation education can adopt a more infection control-friendly format compared to simulation education. Figure 5 shows that non-simulation education can adopt a more infection control-conscious format than simulation education. In other words, our study suggests that appropriate use of simulation and non-simulation education may be able to achieve both educational effectiveness and infection control.

Comment5: You may discuss the best way to teach residents and maintain skills for senior anesthesiologists in regard to yours results.   

Response5: Thank you for your very effective advice. We agree with the reviewer’s comment. Our result shown that that there is no significant difference between Sim and non-Sim in the knowledge score (Kirkpatrick level2). This suggests that replacing a part of medical education with non-Sim, while maintaining good educational effect, may be possible to achieve both educational effect and infection prevention measures. However, the accumulation of medical pedagogical evidence necessary to construct a specific curriculum for this purpose is still insufficient, and future research is desirable. This has been added to the main manuscript.

We are confident that our revised manuscript will be suitable for publication in Journal of Clinical Medicine and look forward to receiving your editorial decision.

Thank you for your consideration.

Sincerely,

Koichi Ando

Department of Medicine, Division of Respiratory Medicine and Allergology, Showa University School of Medicine

1-5-8 Hatanodai, Shinagawa-ku, Tokyo, 142-8666, Japan

Tel: +81-3-3784-8532

Fax: +81-3-3784-8742

Email: koichi-a@med.showa-u.ac.jp